# Tracking miR-17-5p Levels following Expression of Seven Reported Target mRNAs

**DOI:** 10.3390/cancers14112585

**Published:** 2022-05-24

**Authors:** Kevin Y. Du, Javeria Qadir, Burton B. Yang, Albert J. Yee, Weining Yang

**Affiliations:** 1Sunnybrook Research Institute, Sunnybrook Health Sciences Centre, Toronto, ON M4N 3M5, Canada; kevin.yc.du@gmail.com (K.Y.D.); javeriaqadir@hotmail.com (J.Q.); byang@sri.utoronto.ca (B.B.Y.); albert.yee@sunnybrook.ca (A.J.Y.); 2Department of Laboratory Medicine and Pathobiology, University of Toronto, Toronto, ON M5S 1A8, Canada

**Keywords:** microRNA, non-coding RNA, miR-17-5p, 3′-UTR, Argonaute 2, RNA-induced silencing complex, cancer

## Abstract

**Simple Summary:**

MicroRNAs (miRNAs) are non-coding RNA sequences that promote gene silencing by targeting matching mRNAs. miR-17-5p is a typical oncogenic miRNA overexpressed in many types of cancers. Due to imperfect specificity, a single miRNA, such as miR-17-5p, may target multiple mRNAs with a range of tissue-specific effects. Therefore, investigating miRNA functions is rather complex. In this study, miR-17-5p was found to be correlated with and modulated by the tested miR-17-5p downstream target mRNA levels in cancer cell lines, suggesting that these target mRNA levels may play roles in stabilizing and modifying the expression of miR-17-5p. We postulate that the mechanisms regulating miR-17-5p expression by its known target transcripts can provide an understanding of the dysregulated expression and functions of miRNAs in cancer progression.

**Abstract:**

As the most prominent member of the miR-17-92 cluster, miR-17-5p is well associated with tumorigenesis and cancer progression. It can exert both oncogenic and tumor-suppressive functions by inducing translational repression and/or mRNA decay. The complexity of the tissue-specific expression of the targeted transcripts seems to contribute to the differential functions of miR-17-5p in different types of cancers. In this study, we selected 12 reported miR-17-5p targeting genes with mRNA levels unaffected by miR-17-5p expression and analyzed their expression in 31 organ tissues in transgenic mice by real-time PCR. Surprisingly, miR-17-5p expressing transgenic mice showed a positive correlation in these tissues between miR-17-5p expression levels and the selected miR-17-5p targeted transcripts; with high expression of the miRNA in organs with high selected miRNA-targeted mRNA levels. In cancer cell lines, overexpression of 7 reported miR-17-5p targeted genes’ 3′-UTRs promoted miR-17-5p expression; meanwhile, transfection of 3′-UTRs with mutations had no significant effect. Moreover, an increase in AGO2 mRNA was associated with 3′-UTR expression as confirmed by real-time PCR. Hence, miR-17-5p regulation by these target genes might be an alternative mechanism to maintain miR-17-5p expression at tissue-specific levels.

## 1. Introduction

MicroRNAs (miRNAs) are non-coding RNAs (ncRNAs), 18–23 nucleotides in length, involved in gene regulation through post-transcriptional mRNA silencing. By pairing with partial matching mRNAs, miRNAs downregulate the expression of their target genes by inducing translational repression and/or mRNA decay [1]. Targeting is achieved by specific base-pairing interactions between target sites in the 3′ untranslated region (3′-UTR) of mRNA [2], with the incorporation of miRNA into the RNA-induced silencing complex (RISC) leading to subsequent gene silencing. Argonaute proteins are indispensable components of RISC and play a critical part in RNA interference. The Argonaute-2 (AGO2) protein, in particular, has the ability to alter target mRNA translation and initiate cleavage [3]. Perfect or near-perfect complementarity is required for mRNA cleavage, whereas a lower degree of complementarity in the seed region usually results in translational repression [4]. In animals, a lower degree of complementarity is more prevalent [5], therefore, a single miRNA may have multiple mRNA binding sites, affecting various distinct molecular pathways [2]. miRNAs function as epigenetic/endemic regulators of gene expression by repressing target mRNAs, which have been reported to underlie diverse aspects of biology [6]. It is estimated that 40% of the human genome is under post-transcriptional miRNA regulation [7].

The abundance and diversity of miRNAs and their target mRNAs have important implications for cell functions, including metabolism, proliferation, differentiation, cell cycle regulation, apoptosis, and stress responses [8,9,10]. Dysregulation of miRNAs has been linked to developmental abnormalities and human diseases, including cancer [11,12]. miR-17-5p belongs to the miR-17-92 cluster, which is commonly overexpressed in human cancers with known tumor-suppressor pathway targets, such as PTEN and TGFβ [11] and, is therefore termed onco-miR. The miR-17-92 cluster is located on the non-protein-coding gene MIR17HG/C13orf25 in the human genome [13], encoding seven miRNAs: miR-17-5p, miR-17-3p, miR-18a, miR-19a, miR-20a, miR-19b, and miR-92a [14]. Of these, miR-17-5p is characteristic of the cluster, with roles in cell functions and potentially oncogenic properties, and its deficiency has been reported to be lethal in neonatal mice [15]. Studies have also suggested critical tumorigenic roles of miR-17-5p in breast cancer, hepatocellular carcinoma, gastric cancer, and prostate cancer [16,17,18,19]. Elevated levels of miR-17-5p have been noted in solid tumor tissues in liver, gastric, and colorectal cancers, as well as in serum samples, thus strengthening its biomarker potential [17,18,20,21]. Furthermore, the degree of miR-17-5p overexpression may be correlated with disease progression and responsiveness to chemotherapeutics [17,18,21]. Likewise, our previous study showed that miR-17 transgenic mice developed live hepatocellular carcinoma tumors, which were promoted by both miR-17-5p and miR-17-3p [17].

The expression and function of miR-17-5p seem to be tissue-specific and developmental stage-dependent [20]. However, its role in breast, prostate, and lung cancers is not yet fully understood [22]. The complexity of tissue-specific gene expression in miR-17-5p targeting seems to contribute to its multiple facets among different cancers [20]. However, it is unclear whether the expression of miR-17-5p-targeted mRNA could affect miR-17-5p expression and dysregulation in cancers. The effect of miRNA-targeted mRNA levels on specific or global miRNA expression may help explain different interactions that produce tissue-specific effects. Therefore, it is essential to critically examine the existence and nature of these interactions to understand miRNA regulation and functions in cancer development and progression.

## 2. Materials and Methods

### 2.1. Materials

The monoclonal rabbit antibody against Argonaute 2 (AGO2, C34C6, #2897) was purchased from Cell Signaling (Danvers, MA, USA). Horseradish peroxidase-conjugated goat anti-rabbit IgG was obtained from Sigma (SKU 12-348, St. Louis, MO, USA). RNA extract kits (#74034), miRNA RT (#339340), and PCR kits (#203745) were obtained from Qiagen (Hilden, Germany). Western blot membranes were purchased from Bio-Rad (#1620184, Hercules, CA, USA), and detection kits were from MilliporeSigma (#WBKLS0500, Burlington, MA, USA).

### 2.2. Constructs and Primers

Constructs of 3′-UTR sequences of genes, including PTEN, TIMP2, TIMP3, ADCY5, PPAR-α, STAT3, and Fibronectin, were generated in our lab. The miRNA and primer sequences used in this study are listed in Appendix A.

### 2.3. miR-17-5p Transgenic Mice

All mouse experiments were performed in accordance with the NIH Guide for the Care and Use of Laboratory Animals. The animal use protocol was approved by the Animal Care Committee of the Sunnybrook Research Institute. The hsa-miR-17-5p/mmu-miR-17-5p (miR-17-5p) transgenic mouse model was generated by pronuclear microinjection of a miR-17-containing DNA fragment into 57BL/6XCBA, performed using a standard protocol approved by the Animal Use Subcommittee of the University Council on Animal Care, The University of Western Ontario. All transgenic mice were ear-tagged and genotyped after weaning. Genotyping was performed by PCR using primers EGFP-347F pairing with EGFP-668R and EGFP981F pairing with EGFP-CApaI (Appendix A).

In the initial in vivo experiments, we randomly selected 20 miR-17 transgenic mice. Both male and female animals were randomly grouped. Genotyping was performed for all the collected organs. Four mice with the most similar expression levels of the transgene (miR-17) in different organs tested by genotyping. miR-17-5p expression and targeting gene mRNA were analyzed by PCR using these four miR-17 transgenic mice and WT mice.

### 2.4. Cell Culture and Transfection

Human hepatocellular carcinoma cell lines HepG2 and JHH-1, breast cancer cell line MDA-MB-231, prostate cancer cell lines DU145 and PC3, lung cancer cell line A549, and mouse breast cancer cell line 4T1 were grown in Dulbecco’s modified Eagle’s medium (DMEM) (#12491, Thermo Fisher Scientific, Waltham, MA, USA) supplemented with 10% fetal bovine serum (FBS, #26140, Thermo Fisher Scientific, Waltham, MA, USA), 100 μg/mL penicillin/streptomycin (#15140122, Thermo Fisher Scientific, Waltham, MA, USA) at 37 °C in 5% CO_2_ incubator. All cancer cell lines used in this study were purchased from ATCC. They were developed from different tissues and confirmed to be of high transfection efficiency.

Cell transfection was performed with Lipofectamine 2000 (#11668019, Thermo Fisher Scientific, Waltham, MA, USA) following the protocol provided by the company. Two micrograms of plasmids, siRNAs, or miRNAs were used for each ml transfection medium, except if indicated in the figures or figure legends. For co-transfection or multiple transfections, a total amount of 2 µg was used for each ml transfection medium, except if indicated in the figures or figure legends. All cells were transfected for 5 h. After culturing in basal medium for 24 h, the transfected cells were harvested and subjected to RT-PCR.

### 2.5. Western Blotting

For Western blotting, tissues or cells were lysed and subjected to sodium dodecyl sulfate-polyacrylamide gel electrophoresis (SDS-PAGE) containing 7–12% acrylamide. The separated proteins were transferred onto a nitrocellulose membrane in 1× Tris/glycine buffer containing 20% methanol at 60-V at 4 °C for 1.5 h. The membrane was blocked in TBST (10 mM Tris-Cl, pH 8.0, 150 mM NaCl, 0.05% Tween-20) containing 5% non-fat dry milk powder (TBSTM) at room temperature for 0.5 h. The membranes were incubated overnight with 1:2000 primary antibodies against AGO2 at 4 °C. The blots were washed with TBST (3 × 30 min) and incubated with secondary antibodies for 2 h. After being washed with TBST (3 × 30 min), the blots were visualized with an ECL detection kit (#WBKLS0500, Millipore Sigma, Burlington, MA, USA).

### 2.6. Real-Time PCR

Real-time PCR was performed with SYBR Green PCR Kit (Cat# 1725120, Bio-Rad) using 2 μL cDNA as a template with two appropriate primers. For miRNA, real-time PCR was performed using a miScriptSYBR Green PCR Kit (#203745, Qiagen, Hilden, Germany), and the primer specific for mature miRNAs was purchased from Qiagen. Total RNAs were extracted from tissues or cell cultures with the mirVana miRNA Isolation Kit (AM1560, Ambion, Streetsville, ON, Canada) and processed for reverse transcription with the miScript Reverse Transcription Kit from Qiagen (#339340, Mississauga, ON, Canada) using 1 µg RNA. Thermocycler conditions were 32 cycles of denaturation at 95 °C for 15 s, annealing at 55 °C for 10 s, and an extension step of 72 °C for 5 s. All real-time PCR results were calibrated with the ΔΔCT method, in which each relative mRNA value was normalized with small nuclear RNA U6.

### 2.7. Statistical Analysis

All experiments were performed in triplicate or as indicated in the experiments. Data are presented as mean (bar) with standard deviation (SD). Normality of distribution was determined using the Shapiro–Wilk test. Two-tailed unpaired Student’s *t*-test was performed to assess the difference between two groups with a single independent factor. For multiple group analyses, one-way ANOVA followed by Bonferroni post hoc test for one independent variable, and two-way ANOVA followed by Bonferroni correction for two independent variables were performed. Pearson correlation was used to analyze the linear relationship between the two variables. Prism 8 was used for the above statistical analyses, and differences were considered statistically significant when the nominal *p* value was less than 0.05.

## 3. Results

### 3.1. miR-17-5p Expression Positively Correlated with the Selected Target mRNAs in Wild Type and miR-17 Transgenic Mice

MiRNA can downregulate the expression of its target genes by inducing translational repression and/or mRNA decay. Expression of miR-17-5p can decrease the mRNA levels of targeting genes, yielding potentially negative correlations between miR-17-5p and some targeting mRNAs. To investigate whether the expression of targeted mRNAs affects miR-17-5p expression, we analyzed all currently reported miR-17-5p targeting genes whose mRNA levels were not affected by miR-17-5p expression. In this case, the correlation between miR-17-5p and some target mRNAs allowed us to assess the effects of the target mRNA levels on miR-17-5, rather than the effect of miR-17-5p on the levels of the target mRNAs.

The selection amounted to 12 target genes, including Cyclin-dependent kinase inhibitor 1 (p21), Phosphatase and tensin homolog (PTEN), Bcl-2-like protein 11 (bcl2l11), E2F Transcription Factor 1 (E2F1), RB Transcriptional Corepressor 1 (Rb1), RB Transcriptional Corepressor Like 2 (Rbl2), cyclin D1, Nuclear receptor coactivator 3 (NCOA3), P300/CBP-associated factor (PCAF), Connective tissue growth factor (CTGF), Signal transducer and activator of transcription 3 (STAT3) and Adenyl cyclase 5 (ADCY5). These selected genes are targets of miR-17-5p in both mice and humans according to sequence analysis. Target prediction analysis was performed on these targets with the miRTarBase database (https://mirtarbase.cuhk.edu.cn/~miRTarBase/miRTarBase_2022/php/index.php, accessed on 8 February 2022) (Appendix A). The functions of these selected target genes are listed in Appendix A. The effect of miR-17-5p on the expression of these selected target mRNAs was confirmed in mouse breast cancer cell line 4T1 (Appendix A). The expression levels of the 12 selected target mRNAs in 31 organs of wild-type (WT) and miR-17 transgenic mice (TG) were calibrated by real-time PCR (Appendix A). Quantification of the geometrical mean of these 12 target mRNAs is shown in Figure 1a. Expression of miR-17-5p did not change the geometrical mean of the 12 target mRNAs (Figure 1a). miR-17-5p expression was analyzed in 31 organs of the WT and TG mice (Figure 1b). It was obvious that organs with high miR-17-5p expression levels in WT mice also displayed manifold increases in TG mice (Figure 1b). Interestingly, the organs that expressed high miR-17-5p levels in WT and TG mice also displayed a high geometrical mean of selected targeted mRNAs (Figure 1a). The varying expression levels of miR-17-5p in WT mice organs, as well as the increased levels of miR-17-5p in TG mice, are positively correlated with the geometrical mean of the selected target mRNAs (Figure 1c,d), indicating that the selected miR-17-targeted mRNA expression levels may influence miR-17-5p expression in different tissues.

Since only 12 targeting mRNAs were conditionally selected, the correlation analysis in different organs provides an incomplete picture of the association between miR-17-5p and the targeting mRNAs. However, the positive correlation of miR-17-5p and these selected mRNA levels suggests a potential role for the targeting mRNAs in modulating miR-17-5p expression in different organs of mice.

### 3.2. Expression of 3′-UTR Sequences of miR-17-5p Target mRNAs Increased miR-17-5p Levels

To observe how miR-17-5p-targeted mRNAs affect miR-17-5p expression, we transfected human hepatocellular carcinoma cell line HepG2 with coding sequences of miR-17-5p target genes; Phosphatase and tensin homolog (PTEN), Metallopeptidase inhibitor 2 (Timp2), Metallopeptidase inhibitor 3 (TIMP3), Adenyl cyclase 5 (ADCY5), Peroxisome proliferator-activated receptor alpha (PPAR-α), Signal transducer and activator of transcription 3 (STAT3) and Fibronectin (FN). These 7 target genes were randomly selected from reported miR-17-5p targets, which are targets of miR-17-5p in both mouse and human according to sequence analysis. Three of them were in the list of the 12 genes used in Figure 1, and the other 4 were not. Our preliminary study showed that the expression of miR-17-5p decreased the latter 4 genes’ expression at mRNA levels (Appendix A). PCR analysis showed that the expression of these miR-17-5p-target gene coding sequences did not significantly change miR-17-5p expression (Figure 2a). We generated 3′-UTR and 3′-UTR with mutated miR-17-5p binding sites (UTR-M) constructs of these 7 target mRNAs and found that overexpression of 3′-UTR sequences of these miR-17-5p target genes, but not mutated 3′-UTR, significantly increased miR-17-5p expression in HepG2 cells (Figure 2b). Specifically, the 3′-UTR of the tumor-suppressor gene PTEN significantly promoted miR-17-5p expression in a dose-dependent manner (Figure 2c). However, PTEN-UTR with miR-17-5p binding site mutation construct (PTEN-UTR-M) mitigated the effect of PTEN 3′-UTR on miR-17-5p expression (Figure 2c). There was a positive correlation between fold increases of PTEN 3′-UTR and miR-17-5p in the PTEN 3′-UTR transfected HepG2 cells (Figure 2d).

miR-17-5p was co-transfected with control vector, 3′-UTRs or 3′-UTR with mutated miR-17-5p binding sites of PTEN, Timp2, Timp3, ADCY5, PPAR-α, STAT3, and FN or a control vector in human HepG2 cells. miR-17-5p and above targeted mRNA 3′-UTR sequences co-transfected cells expressed significantly higher levels of miR-17-5p than cells co-transfected with a vector construct or mutated 3′-UTR (Figure 3a). HepG2 cells co-transfected with miR-17-5p and 3′-UTR sequences of PTEN expressed much higher levels of miR-17-5p than the control group samples, which was dose-dependent and could be prevented by mutation in miR-17-5p binding sites (Figure 3b). There was a positive correlation between PTEN 3′-UTR overexpression and miR-17-5p in the miR-17-5p and PTEN 3′-UTR co-transfected HepG2 cells (Figure 3c). The effect of the expression of PTEN 3′-UTR on miR-17-5p expression in miR-17-5p transfected cells was confirmed in the breast cancer cell line MDA-MB-231, prostate cancer cell line DU145, and lung cancer cell line A549. All cell lines demonstrated that 3′-UTR sequences of PTEN, but not the coding sequences, increased miR-17-5p expression (Figure 3d and Appendix A). The direct binding of the PTEN 3′-UTR sequence and miR-17-5p may be integral to the modulatory process, as mutation in miR-17-5p sites eliminated the effect of target mRNA on miR-17-5p expression (Figure 3d).

### 3.3. Expression of miR-17-5p and Selected Target mRNAs Enhanced AGO2 Expression

miRNAs downregulate the expression of their target genes by binding to 3′-UTR and destabilizing mRNA or inducing translational repression. This process occurs via RNA-induced silencing complexes (RISCs), which contain the Argonaute subfamily protein as a core component. Argonaute 2 (AGO2) acts as a crucial contributor to the gene-silencing function of the complex. We analyzed AGO2 expression in HepG2 cells co-transfected with miR-17-5p and 3′-UTR sequences or 3′-UTR-M of PTEN, Timp2, Timp3, ADCY5, PPAR-α, STAT3, or FN, and compared these with co-transfected miR-17-5p and a control vector. Selected miR-17-5p-targeted 3′-UTR, but not 3′-UTR-M transfected cells, showed increased AGO2 mRNA (Figure 4a). To confirm the effect of PTEN 3′-UTR (PTEN-UTR) expression on AGO2 expression, PTEN-UTR and PTEN-UTR-M were transfected into human cancer cell lines HepG2, MDA-MB-231, DU145, and A549. PCR results showed that expression of PTEN-UTR, but not PTEN-UTR-M increased AGO2 expression at mRNA levels (Figure 4b).

To screen for tissue-specific effects, AGO2 mRNA levels were measured in 31 organs from the WT and TG mice. It seemed that those organs that expressed high levels of AGO2 mRNA in WT mice also had significantly increased AGO2 mRNA levels in TG mice (Figure 4c). A positive correlation was observed between AGO2 mRNA and miR-17-5p in the organs of TG mice (Figure 4d and Appendix A). The fold increase of AGO2 mRNA was also positively correlated with the geometrical mean of targeted mRNAs in these 31 organs of TG mice (Figure 4e). Though the expression of PTEN 3′-UTR increased AGO2 expression at mRNA levels, there seemed to be no significant change in AGO2 expression at protein levels with blotting (Figure 4f). The AGO2 protein also exhibited similar expression in liver, lung, and prostate tissues in WT and miR-17-5p transgenic mice (Figure 4g). The above results showed that the expression of miR-17-5p or miR-17-5p-targeted mRNAs increased the expression of AGO mRNA but had no significant effect on AGO protein levels.

Confirmation of AGO2 expression with miR-17-5p and miR-17-5p-targeted mRNA was performed with hepatocellular carcinoma cell lines HepG2 and JHH-1, breast cancer cell lines MDA-MB-231, prostate cancer cell lines DU145 and PC3, and lung cancer cell line A549 (Figure 5a). A positive correlation was detected between AGO2 mRNA and miR-17-5p in above miR-17-5p transfected cells (Figure 5b). Another PCR analysis was performed with HepG2 cells transfected with miR-17-5p, miR-17-3p, miR-24, miR-93, miR-98, miR-199a-3p, miR-299-3p, miR-330-3p, miR-378, miR-491-5p, and miR-661. Expression of miR-17-5p, miR-17-3p, miR-24, miR-93, miR-98, miR-199a-3p, miR-299-3p, miR-330-3p, miR-378, miR-491-5p, and miR-661 enhanced AGO2 expression (Figure 6a). A positive correlation was observed between the fold increase of AGO2 mRNA and miRNA in above miR-17-5p, miR-17-3p, miR-24, miR-93, miR-98, miR-199a-3p, miR-299-3p, miR-330-3p, miR-378, miR-491-5p, and miR-661 transfected cells (Figure 6b). These results, together, indicated that expression of miR-17-5p, as well as other tested miRNAs, increased AGO2 mRNA expression levels.

## 4. Discussion

Dysregulation of miR-17-5p occurs frequently in cancers, with studies indicating primarily oncogenic effects. Elevated miR-17-5p expression was noted in many cancer types, and the degree of overexpression could be correlated with disease progression and poor prognosis [17,19,21,23]. However, in some cancer types, such as lung, breast, cervical, and prostate cancer, the function of miR-17-5p is not nearly as obvious. Despite being labeled as an oncomiR, a tumor-suppressive role has also been recorded in several studies. The fact that miR-17-5p exerts differential functions implicates the complexities of cancer progression, as well as the intricacies of the regulatory network of miRNAs [7]. It is well known that one miRNA may regulate different target genes [24]. The complexities of miRNA functions may be partly attributed to imperfect matching, with only a few matching nucleotides necessary for miRNA/mRNA binding. Based on this, the association of miRNA RISC results in the repression of the target gene by promoting mRNA translational inhibition. The imperfect nature of the miRNA–mRNA interaction implies that a single miRNA could potentially target and exert effects on tens to hundreds of mRNAs involved in many distinct pathways, which may perform opposing functions [23,25]. Conversely, this also means that a single mRNA may be targeted by many miRNAs. The expression and functions of miR-17-5p are tissue-specific as observed by varying levels in different mouse organs. This phenomenon has mainly been attributed to transcriptional regulation. However, increasing evidence supports the regulation of mammalian miRNA expression at the post-transcriptional level. miRNAs may be spatially expressed in distinct cell types, while there are precursors ubiquitously expressed throughout all the tissues analyzed, indicating post-transcriptional modulation as an alternative mechanism to control miRNA expression and function [26]. Expression of miR-17-5p could repress target gene translation or enhance target mRNA degradation. To investigate whether the expression of targeted mRNAs affects miR-17-5p expression, we investigated currently reported miR-17-5p target genes, finding 12 with mRNA levels unaffected by miR-17-5p expression. A positive correlation between miR-17-5p expression levels and the selected miR-17-5p targeted gene transcript levels in 31 mouse organs suggested that tissue-specific expression of target mRNAs may modulate miR-17-5p expression at post-transcriptional levels. We then randomly selected 7 reported miR-17-5p target genes. Expression of miR-17-5p decreased 4 of the 7 target genes on mRNA levels, which were not in the selected 12 genes. Interestingly, we found that the increased presence of 3′-UTR sequences of these miR-17-5p target mRNAs through transfection promoted miR-17-5p expression in cancer cell lines from different tissues. While expression of 3′-UTR of the 7 genes significantly enhanced miR-17-5p expression, this was prevented by mutation of miR-17-5p 3′-UTR binding site. Clarification of the mechanisms of the regulation of miR-17-5p by its target mRNA 3′-UTR in mice tissues and cancer cell lines may offer insight into the regulation and differential oncogenic effects of miRNA.

The recent identification of the mechanisms of miRNA biogenesis regulation uncovers that various factors or growth factor signaling pathways control each step of the miRNA biogenesis pathway [27]. Characterization of the regulatory elements of miRNA biosynthesis and function will provide new insights, yielding a comprehensive understanding of the complex gene regulatory networks governed by miRNAs and the involvement of miRNAs in various pathological mechanisms [28]. Several studies have demonstrated that miRNA maturation pathways crosstalk with intracellular signaling molecules, including p53, Smad proteins, and estrogen receptors [29,30,31,32]. Multiple RNA binding proteins have been demonstrated to be involved in the biased processing of different miRNA species [33]. Recently, a novel paradigm for mRNA function independent of protein coding was identified. Messenger RNAs, including pseudogene mRNAs, bind targeting miRNAs, and thus alleviate miRNA-mediated repression on mRNAs whose 3′ UTRs have a common miRNA binding site [34]. Genetic and genomic evidence suggests that pseudogenes of oncogenes and tumor suppressor genes have a significant impact on cancer biology [35,36]. Thus, due to the effect of target-directed protection or degradation on miRNAs, it is conceivable that pseudogene mRNAs may stabilize or destabilize their targeting miRNAs besides acting as a competitor between miRNAs and protein-coding genes [37]. However, this cannot fully explain the regulation of miRNA levels by its targeted mRNA, and how miR-17-5p-binding mRNA can increase miR-17-5p levels.

Our previous studies showed that the dynamic binding of miRNAs with 3′-UTR was important in modulating miRNAs’ function [38,39]. In this study, we found that miR-17-5p expression levels were also regulated by 3′-UTRs of 7 known target mRNAs. Taken together these results with previous publications, a possible explanation is that the binding of miR-17-5p with its target 3′-UTR mRNA may play a role in maintaining its stability. During gene silencing by miRNA, the RISC containing miRNA and AGO2 forms a dynamic complex with the 3′-UTR of the target mRNA, which results in repressing mRNA translation. Expression of miRNA and 3′-UTR of target genes, as well as sufficient AGO2, is necessary to maintain the complex and inhibit target gene translation. The formation of this complex may also be important in stabilizing miRNA and reducing its degradation (Figure 7). The dynamic binding of miR-17-5p with these target 3′-UTRs may be important to protect miR-17-5p from target-directed miRNA degradation. Expression of miR-17-5p or its target 3′-UTR increased AGO2 mRNA levels, which can be a response to decreased free AGO2. However, increased AGO2 mRNA levels were not associated with changes in total AGO2 protein levels using western blots in either cancer cell lines and miR-17 TG mice, implicating post-transcriptional regulation to stabilize protein levels. Silencing interactions of different miRNAs are dependent on and regulated by interactions with specific Argonaute proteins, with varying Argonaute protein ratios in different tissues [40].

## 5. Conclusions

Expression levels of mRNA may be able to regulate targeting miRNA expression, with correlating levels among different tissues. The presence of miR-17-5p in silencing complexes, preventing its degradation in response to increased target mRNA 3′-UTRs may be an alternative mechanism to control miRNA expression, warranting ongoing study.

## Figures and Tables

**Figure 1 cancers-14-02585-f001:**
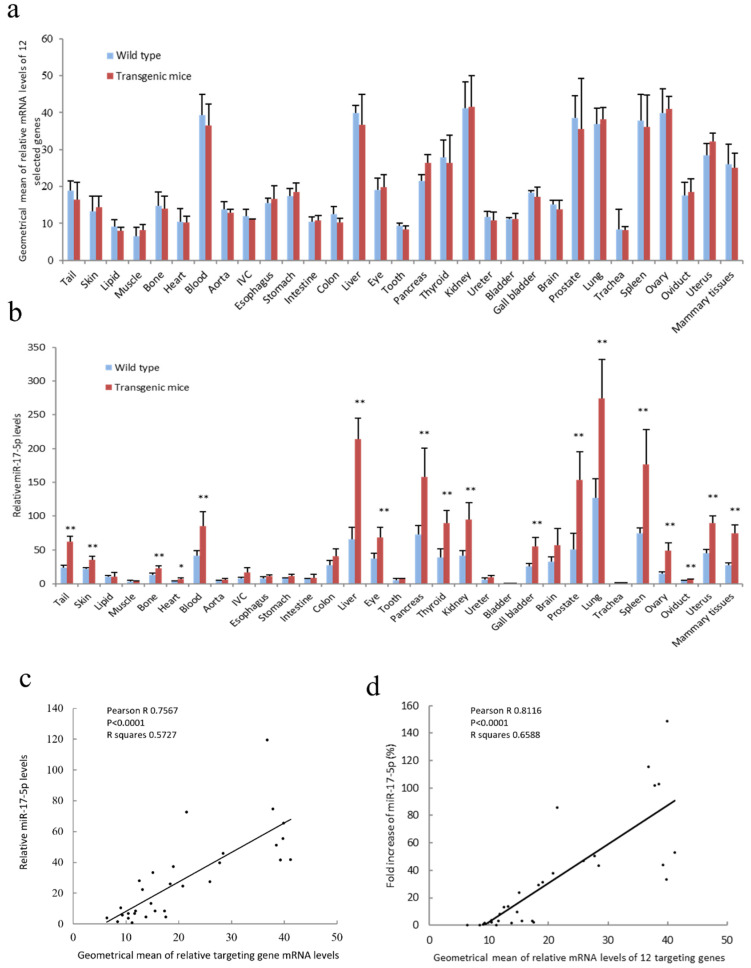
miR-17-5p expression was positively correlated with the selected target mRNAs in transgenic mice. (**a**) Total RNAs were collated from tissues of 31 different mouse organs and subjected to reverse transcription and real-time PCR using primers against 12 selected miR-17-5p target genes, including p21, PTEN, bcl2l11, E2F1, Rb1, Rbl2, cyclin D1, NCOA3, PCAF, CTGF, STAT3, and ADCY5. Bars show the geometrical means of these miR-17-5p target mRNAs in wild-type and miR-17 transgenic mouse tissues. Error bars, SD; *n* = 4. (**b**) Mouse organ tissues were subjected to PCR, showing that miR-17 transgenic mice expressed increased miR-17-5p levels compared to WT mice. The expression levels of miR-17-5p are different in different organ tissues. * *p* < 0.05, ** *p* < 0.01 vs. WT; Error bars, SD; *n* = 4. (**c**) Pearson correlation of miR-17-5p expression levels and the geometrical mean of 12 miR-17-5p targeted mRNAs in WT mouse organs was analyzed using Prism 8. A positive correlation was uncovered between the miR-17-5p expression levels and the geometrical mean of these 12 miR-17-5p target mRNAs in mouse tissues. Pearson R = 0.7567; *p* < 0.0001; *n* = 31. (**d**) Pearson correlation of fold increases of miR-17-5p and the geometrical mean of 12 miR-17-5p target mRNAs was analyzed by Prism8. Pearson R = 0.8116; *p* < 0.0001; *n* = 31.

**Figure 2 cancers-14-02585-f002:**
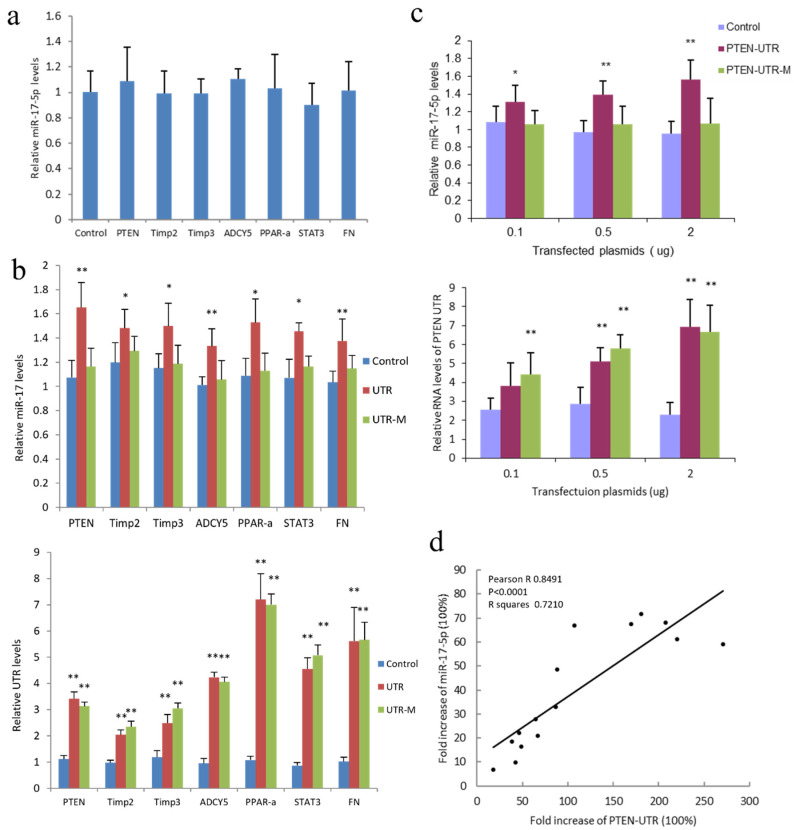
Expression of PTEN 3′-UTR increased miR-17-5p levels. (**a**) Hepatocellular carcinoma cell line HepG2 cells were transfected with plasmids of coding sequences of miR-17-5p target genes PTEN, Timp2, Timp3, ADCY5, PPAR-α, STAT3, and Fibronectin (FN), and subjected to RT-PCR. Expression of these miR-17-5p target gene coding sequences did not significantly change miR-17-5p expression levels. (**b**) Upper, HepG2 cells were transfected with plasmids with 3′-UTR sequences (UTR) of above miR-17-5p targeting genes and 3′-UTR with mutated miR-17-5p binding sites (UTR-M), then subjected to RT-PCR. Expression of 3′-UTR sequences of these miR-17-5p target genes, but not 3′-UTR with mutated miR-17-5p binding sites, significantly increased miR-17-5p expression. Lower, the expression of the above 3′-UTR sequences in the transfected cells. * *p* < 0.05, ** *p* < 0.01 vs. control; Error bars, SD; *n* = 5. (**c**) Upper, HepG2 cells were transfected with a plasmid with PTEN 3′-UTR (PTEN-UTR) and PTEN 3′-UTR with mutated miR-17-5p binding sites (PTEN-UTR-M) in varying concentrations. PCR showed that expression of PTEN-UTR, but not PTEN-UTR-M increased miR-17-5p levels, which was dose-related. Lower, the expression of PTEN 3′-UTR in the transfected cells. * *p* < 0.05, ** *p* < 0.01 vs. control; Error bars, SD; *n* = 5. (**d**) Pearson correlation between fold increases of PTEN UTR and miR-17-5p in the PTEN UTR-transfected HepG2 cells was analyzed by Prism 8. Fold increases of PTEN 3′-UTR were positively correlated with increasing miR-17-5p levels. Pearson R = 0.8491; *p* < 0.0001; *n* = 15.

**Figure 3 cancers-14-02585-f003:**
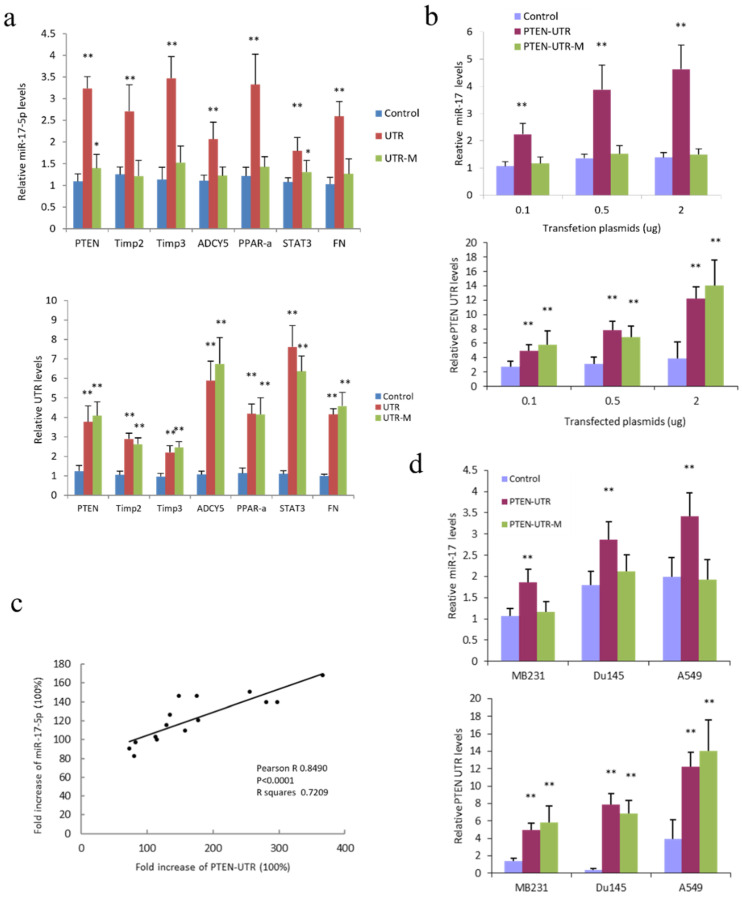
Increased expression of miR-17-5p target gene 3′-UTR increased miR-17-5p levels (**a**) Upper, HepG2 cells were co-transfected with miR-17-5p and a control vector, 3′-UTR or 3′-UTR with mutated miR-17-5p binding sites (UTR-M) of PTEN, Timp2, Timp3, ADCY5, PPAR-α, STAT3, and FN. 3′-UTR transfected cells showed increased miR-17-5p levels, but the UTR-M transfected cells showed comparable miR-17-5p levels to control. Lower, PCR showed the expression of 3′-UTR of the above genes in the transfected cells. * *p* < 0.05, ** *p* < 0.01 vs. control; Error bars, SD; *n* = 5. (**b**) Upper, HepG2 cells were co-transfected with miR-17-5p (1 µg/mL) and a control vector, PTEN-UTR or PTEN-UTR-M. Expression of PTEN-UTR, but not PTEN-UTR-M increased miR-17-5p levels, which was dose-related. Lower, the expression of PTEN 3′-UTR in the transfected cells. ** *p* < 0.01 vs. control; Error bars, SD; *n* = 5. (**c**) Pearson correlation between fold increase of PTEN UTR and miR-17-5p in the PTEN UTR and miR-17-5p co-transfected HepG2 cells was analyzed by Prism 8. Fold increases in PTEN 3′-UTR were positively correlated with miR-17-5p. Pearson R = 0.8490; *p* < 0.0001; *n* = 15. (**d**) Upper, miR-17-5p was co-transfected with control vector, PTEN-UTR or PTEN-UTR-M in breast cancer (MDA-MB-231), prostate cancer (DU145), and lung cancer cell lines (A549). The expression of PTEN-UTR increased miR-17-5p levels in all cell lines. Lower, expression of PTEN 3′-UTR in the transfected cell lines. ** *p* < 0.01 vs. control; Error bars, SD; *n* = 5.

**Figure 4 cancers-14-02585-f004:**
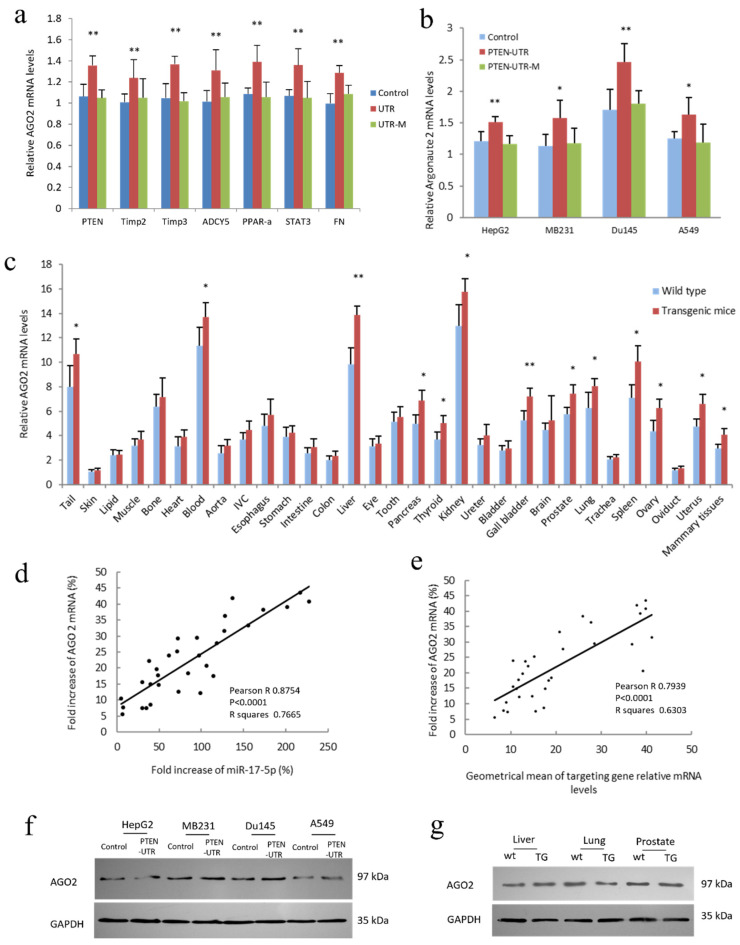
AGO2 mRNA increased in PTEN-UTR transfected cancer cell lines and miR-17-5p transgenic mice. (**a**) HepG2 cells were co-transfected with miR-17-5p and a control vector, 3′-UTR or 3′-UTR-M of PTEN, Timp2, Timp3, ADCY5, PPAR-α, STAT3, and FN. 3′-UTR transfected cells showed increased AGO2 mRNA levels. ** *p* < 0.01 vs. control; Error bars, SD; *n* = 5. (**b**) miR-17-5p and a control vector, PTEN-UTR or PTEN-UTR-M were co-transfected to HepG2, MDA-MB-231, DU145, and A549 cell lines. Expression of PTEN-UTR increased miR-17-5p levels in these cell lines. * *p* < 0.05, ** *p* < 0.01 vs. control; Error bars, SD; *n* = 5. (**c**) miR-17-5p transgenic mice expressed increased AGO2 mRNA levels compared to WT mice. Expression varies among organs. * *p* < 0.05, ** *p* < 0.01 vs. WT; Error bars, SD; *n* = 4. (**d**) Pearson correlation between fold increases of AGO2 mRNA and miR-17-5p in organs was analyzed by Prism 8. Fold increases in AGO2 mRNA were positively correlated with miR-17-5p. Pearson R = 0.8754; *p* < 0.0001; *n* = 31. (**e**) Pearson correlation between fold increases of AGO2 mRNA and geometrical mean of 12 cancer-related miR-17-5p target mRNAs in organs was analyzed by Prism 8. The fold increase of AGO2 mRNA was positively correlated with the geometrical mean of these 12 miR-17-5p target mRNAs. Pearson R = 0.7939; *p* < 0.0001; *n* = 31. (**f**) HepG2, MB231, Du145, and A549 cells were co-transfected with miR-17-5p and a control vector or 3′-UTR of PTEN and subjected to western blot with an antibody against AGO2. Expression of PTEN-UTR did not noticeably change AGO2 at protein levels. (**g**) Liver, lung, and prostate tissues from WT and miR-17-5p transgenic mice were lysed and subjected to western blot with an antibody against AGO2. Increased expression of miR-17-5p in transgenic mice did not noticeably change AGO2 expression at the protein level. Original blots see Appendix A.

**Figure 5 cancers-14-02585-f005:**
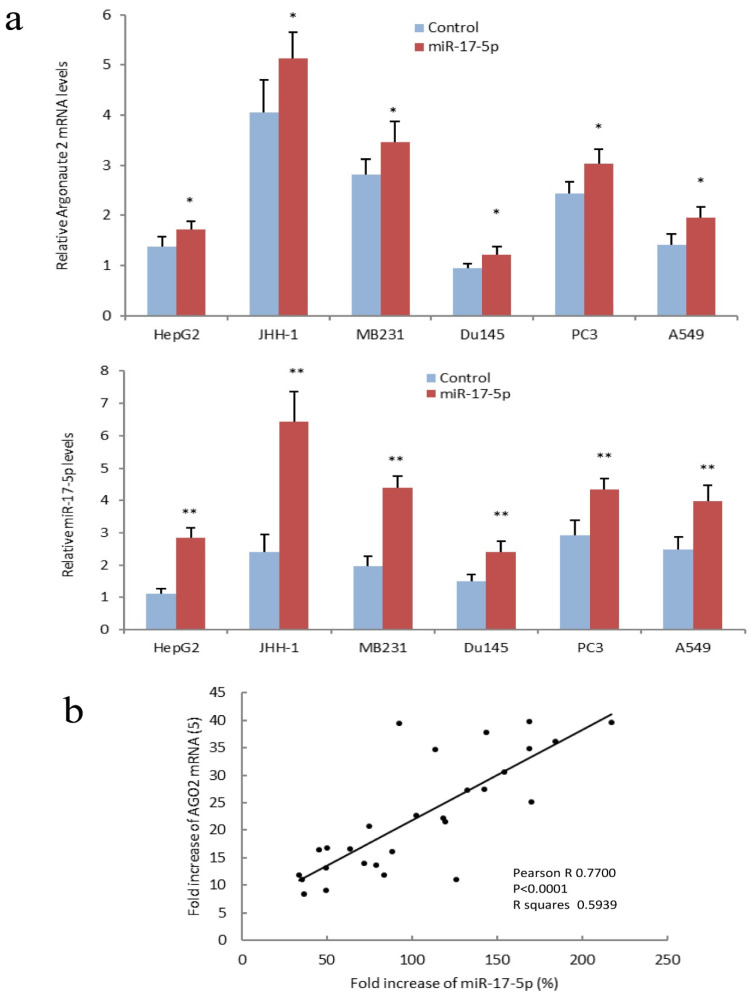
AGO2 expression increased in miR-17-5p transfected cancer cell lines. (**a**) Upper, RT-PCR showed that AGO2 expression increased in miR-17-5p transfected HepG2 cells. Lower, expression of miR-17-5p expression in above-transfected cell lines. * *p* < 0.05, ** *p* < 0.01 vs. control; Error bars, SD; *n* = 5. (**b**) Pearson correlation between fold increase of AGO2 mRNA and miR-17-5p in above miR-17-5p transfected cells was analyzed by Prism 8. Fold increases in AGO2 mRNA were positively correlated with increased miR-17-5p levels. Pearson R = 0.7700; *p* < 0.0001; *n* = 28.4.

**Figure 6 cancers-14-02585-f006:**
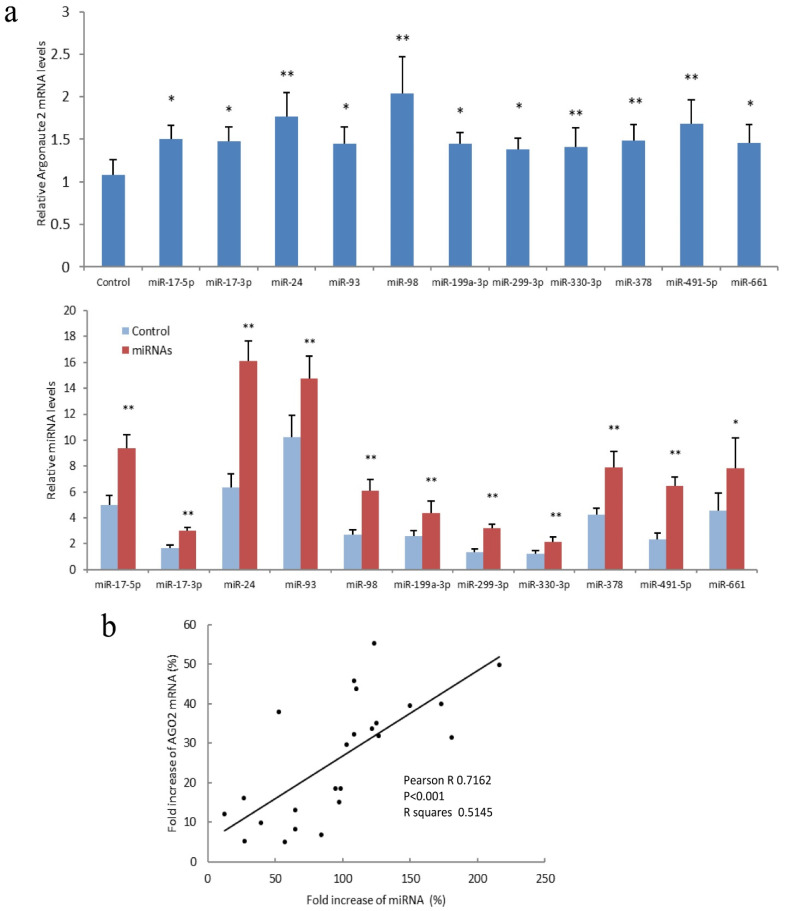
AGO2 expression increased with miRNA transfection. (**a**) Upper, HepG2 cells were transfected with miR-17-3p, miR-17-5p, miR-24, miR-93, miR-98, miR-199a-3p, miR-299-3p, miR-330-3p, miR-378, miR-491-5p, and miR-661. RT-PCR showed that AGO2 expression increased with the above miRNA transfection. Lower, expression of miR-17-3p, miR-17-5p, miR-24, miR-93, miR-98, miR-199a-3p, miR-299-3p, miR-330-3p, miR-378, miR-491-5p, and miR-661 expression in transfected HepG2 cells. * *p* < 0.05, ** *p* < 0.01 vs. control; Error bars, SD; *n* = 5. (**b**) Pearson correlation between fold increase of AGO2 mRNA and fold increase of miR-17-3p, miR-17-5p, miR-24, miR-93, miR-98, miR-199-3p, miR-299-3p, miR-330-3p, miR-378, miR-491-5p, and miR-661 in transfected HepG2 cells was analyzed by Prism 8. Fold increases in AGO2 mRNA were positively correlated with miRNA levels. Pearson R = 0.7162; *p* < 0.001; *n* = 44.

**Figure 7 cancers-14-02585-f007:**
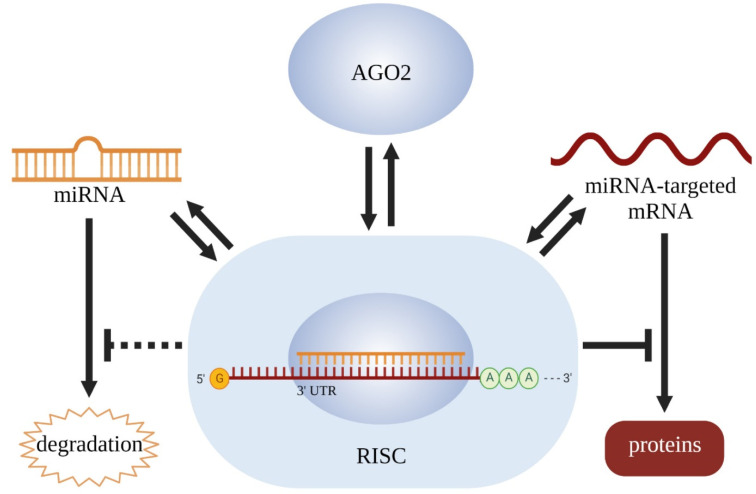
Diagrammatic representation of observed relationships between tissue-specific miRNA and targeted mRNA expression.

## Data Availability

The original data presented in this study are openly available upon request.

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
