# Peer review of "Tracking miR-17-5p Levels following Expression of Seven Reported Target mRNAs"

_cancers, 2022, doi:10.3390/cancers14112585_

Round 1

Reviewer 1 Report

The submitted article entitled “Tissue-specific miR-17-5p target genes levels modulate miR-17-5p expression” aims at showing that the expression of a mature microRNA, miR-17-5p, is controlled by the expression of a few of its targets in human cancer cell lines, and that a few of the observations remain in mice organs.

This article presents interesting figures showing the positive correlation between a miR expression and the expression of a few of its targets. Overall, the positive correlations are quite high and impressive. As this fact is clearly counter-intuitive (miRs are known to repress their targets), the proofs provided should definitely be more convincing and reinforced. Additionally, the article lacks important technical details and requires at least major changes before being publishable. However, the facts shown here seem interesting and deserve further investigations.

An important point is that the proofs provided are often insufficient for the title of the article and of various paragraphs. With the figures provided, another possible explanation is that the transfected RNA (3’UTR) “sequester” the miRNA and compete with another target performing target-directed miRNA degradation (TDMD, see https://www.science.org/doi/full/10.1126/science.abc9359 and references therein). Such phenomenon should be experimentally checked and further discussed.

An important point to be improved is the clarity of the article. It should be clearly stated that what is shown is surprising, as we would expect negative correlation. Thus, a clear caution in the explanations should appear to avoid confusions for the readers. Also, the organism of interest (mouse or human) is not clear throughout the article, including whether mmu-miR-17-5p or hsa-miR-17-5p are considered (and their targets).

Important technical details are missing. It includes the number of mice in each experiment, what is the transgenic mouse (at least minimal information without requiring reading ref 18), the concentration of the transfected miRNAs and RNAs together with a comparison with endogenous levels. In addition, what is the mutation in PTEN-UTR-M (please provide the sequence together with the mature miR and the Watson-Crick pairings)?

Other major comments:

  • As the message insists on the importance of the results in cancer, an easy confirmation would be to check these positive correlations in human tumor tissues through public databases (TCGA for example)
  • The mechanism at stake should be investigated
  • Introduction: references should be all checked. They are often not the proper ones, nor the most updated ones.
  • Paragraph 3.1: I disagree with the sub-title and the message provided. miR-17 is increased in transgenic mice, so it is not controlled by targets, but it may control them (this is miR-17 which is affected by the transgenic mouse). Feedback loops are possible, but should then be further demonstrated.
  • The choice of random targets in not rational. Why random? Why these targets? These targets are definitely not the most probable ones looking at TargetScan prediction for example. I would suggest to provide a transcriptomic profile for a few organs and to provide a convincing explanation of your choice.
  • Paragraph 3.2: why 8 target genes and not 12 as previously? How they were chosen?
  • What does ug of plasmids means in terms of concentration? And compared with endogenous levels? With co-transfection, what is the concentration of each? With multiple transfections (mixtures of miRNAs), what are the individual concentrations / how is done the mixture?
  • 4b, 4d: there is a co-transfection with miR-17-5p, so why there is no increase of miR-17-5p when PTEN-UTR-M is used?
  • l353: clearly overstated. It requires additional proofs.

Minor comments:

  • Paragraph 2.4 which quantity of control RNA is used?
  • [ref] l113 (to remove or replace?)
  • Paragraph 3.3 title: “its target” should be replaced by “selected targets”
  • l250-251, also l322: hard to understand, please rephrase
  • Check miRNA names. “*” should not be used anymore (replace by -3p or -5p). And check the missing dashes “-“
  • l333: ref 29 is not recent (2010).

Author Response

Thank you for your careful review of our manuscript and your suggestions for modification and correction. We have worked carefully to address the comments in the rebuttal letter . 

Reviewer 2 Report

This study explored the phenomenon of miR-17-5p downstream target genes regulating the expression of miR-17-5p itself at the animal level. As the abundance of target genes varies between tissues, the expression of these target genes can in turn regulate the expression of miR-17-5p. The data from this study are also able to support the authors' findings. However there are several questions that require further answers from the authors.

  1. The authors do not give information on the primer probes for the genes tested in the Materials and Methods section, which may result in the authors' findings not being reproduced.
  2. The authors randomly selected 12 miR-17-5p downstream target genes, which are all oncogene, but the authors did not mention in the article from which genes the 12 target genes were selected. In addition, it is recommended that the authors provide information on all target genes predicted by miR-17-5p and the function of the target genes.
  3. In line205, the authors mention PTEN-UTR-M, but the authors do not provide information on the mutation site, nor do they verify whether this mutation site affects the binding efficiency to miR-17-5p. If the authors got this from a reference, please provide the reference.
  4. The authors used the indicator foldincrease in Figure 4. The exact calculation of this indicator is not provided by the authors. Also, the authors should have used the spearson correlation to describe the correlation between these two variables.
  5. Line293, the same miRNA provided by the authors without primer information.

Author Response

(The authors gave the same response as above.)

Reviewer 3 Report

The manuscript analyses the role of the microRNA miR-17-5p, an oncogenic miRNA usually overexpressed in several cancers. Authors assessed the correlation of this miRNA levels with its targets’ mRNA expression and the possible feed-back regulatory loop. To this end, they used different approaches for checking this regulation in various cancer cell lines and mouse tissues.

The results of this study are well presented and shown in figures, that help to better visualise them. However, I would like to address the following comments to this manuscript:

Major comments:

  1. Line 23. Abstract. Authors should add not only the models that they used for the project as they mentioned, but also the methodology (i.e., “…expression on miR-17-5p activities and dysregulation in mouse tissue and cancer cell lines by means of…” and cite the main techniques).
  2. Line 35. Introduction. Authors described miRNAs as “18-23 nucleotides long non-coding RNAs”. I suggest to either substitute “long” for “length” or change it for “short” since the sentence could be confusing and made think that they are included in the “long non-coding” RNAs category.
  3. Materials and Methods. Please, provide the catalogue number or reference of the kits and products indicated in this section.
  4. Please, provide a brief description of the cell lines that have been used in the manuscript and origin (e.g., features, why authors chose these ones and not others). Please, do the same for the mice model and their tissue (any other material) extraction, storage and processing steps.
  5. Line 130. Results. I would like to know the rationale for selecting 12 random targets for miR-17-5p and making an average of their expression value for each organ and later on to check the same in the transgenic mice. miRNAs have demonstrated to have a high tissue specificity and to perform the average of expression of these 12 targets only cover the specific changes on expression that could be happening in some of them. Please, provide the explanation for doing so, or separate the expression of these target genes to confirm that there is no significant change in their expression levels when you express miR-17-5p in 31 organs. This changes also apply to figure 1a). Figure 1 b) also shown that the expression levels of miR-17-5p didn’t change in several organs between WT and TG mice so you wouldn’t expect any change in the expression levels of their target genes. Authors could separate this information and move it to the supplementary information, making the figure clear and more informative to the readers.
  6. Line 151. Some images need to be improved as Figure 1 c) and d). Please, provide images with higher quality if possible. In these figures, authors don’t need to include “Graph Prism 8” or Number of XY, it doesn’t provide any important information in the image and it’s already included in the figure legend and Material and methods section. Please, carefully review the figure legend for this figure. Same comments apply to Figure 2 d) (Line 197), Figure 3 c) (Line 211), Figure 4 d) and e) (Line 225), Figure 5 b) (Line 284) and Figure 6 b) (Line 291).
  7. Line 267. I would like to understand the hypothesis behind the experiments explained in this section, in which authors make a transfection with a mixture of 11 miRNAs, a second mixture of another 11 miRNAs etc. Please, provide information about the selection criteria for these miRNAs’ mixtures, add an explanation of what are you intended to observe and assess performing this, the list of miRNAs included in the mixture (not only in the text but also include a table with all the information related in supplementary information).
  8. Line 292. Figure 6 legend. In line with the previous comment, the information contained in the figure legend related to the mixture of miRNAs need to be more specific, and please don’t include confusing things like “in the above transfected cell lines” (line 296), just include the names of the cell lines for make it easy to read and understand.

Author Response

(The authors gave the same response as above.)

Round 2

Reviewer 1 Report

In this updated version (v2), authors have clarified a few points, and modified the text in a moderate way. While the draft has improved its quality, I do think further improvements are still needed.

If I understand correctly, 12 target genes have been selected first (in mice, which should appear clearer), then 8 target genes have been selected randomly, 3 of them belonging to the previous list (now in human cell lines, which again should appear clearly). Suggestion: for each paragraph, specify the organism of interest. These two selections really need more justifications:

  • The 12 mouse genes have been selected with preliminary experiments to be not affected by miR-17-5p. I do not understand the rational, and I do not see these results in the present work. The justification of this choice clearly deserves a dedicated paragraph with results and justification of the strategy.
  • The 8 human genes are chosen randomly. 3 of them belongs to the previous list. How can we believe that these genes are chosen randomly as only 2 genes belonging to the previous list has a probability of around 0.0008 (if we consider the 1385 conserved targets predicted by TargetScan)! (3 genes have a probability < 1e-5)…

So, an important point is to clearly justify the choice of the target genes chosen, with additional figures and justifications.

Besides, to further reduce overstatement, an easy improvement is to simply change “its targets” by “a few selected targets” everywhere it is necessary (l11, 28, etc.).

An important point I had not sufficiently understood in the first version, is that the correlation shown Fig. 1 is across tissues. It means that the selected targets are more expressed in tissues for which miR-17-5p is also more expressed. This should appear more clearly in the text. Thus, the correlation observed Fig. 1 may be due to co-expression of different members of the same pathway(s) (here miR-17-5p and a few of its “targets”), at least partly.

Importantly, while the experiments showing the transfection of a few genes UTR (and the mutated version of PTEN) are really interesting, we would need to see whether the same phenomenon remains true for the other targets. A particularly interesting case would be to mutate only one of the sites, and all, and measure the differences. Finally, Suppl. Fig. 2 (target sites) should be provided also for the 8-gene set. To note, Suppl. Fig. 2 mentions hsa-miR-17-5p targets, while these are the mice targets used in Fig. 1. Please check.

I do not understand the rational of the co-transfection experiment (Fig. 3). This deserves more and/or clearer justification.

Additional comments:

  • In all y-axis labels, “relative” concentration should be updated in each case with the reference used to normalize (either small nuclear RNA U6 or GAPDH), and the quantity added of these normalizing molecule should be always provided (l148 for example).
  • Paragraph 2.4. When were the measurements performed after transfection?
  • L147/148: deserve more precision
  • L166: Deserves a paragraph. Show it and explain the rational.
  • L178/180: I guess this is Fig. 1b
  • L215: please, show it
  • L216/234: remove space for miR-17 -5p / co- transfected
  • L237 “but not the coding sequence”: please, show it
  • L237: not “target mRNAs” but only PTEN. Please, again, correct everywhere it is necessary, to avoid unproven generalization (this can be raised in the discussion as an hypothesis)
  • 2 legend: It should be completely reviewed to be clearer (and maybe shorter: l244 should be in the main text for example). I do think the main title should be “Expression of miR-17-5p following transfection of coding or 3’UTR sequence of PTEN.” Is this correct? Please, improve the quality of all figure legends accordingly.
  • 4f: not only HepG2 cells
  • L289: I would write “destabilizing” instead of “cleaving” (it is more often through decapping than cleaving itself).
  • L315: one does not see any transfection of 21 microRNAs. Which ones? Results?
  • L373: targeting -> target
  • L375: indicated -> suggested

Author Response

We appreciate the reviewers’ constructive suggestions and comments and have followed the suggestion for revising the manuscript.  We have responded to the academic editor and reviewers on a point-by-point basis to comments raised. Please see attached file.

Reviewer 3 Report

The authors have completed the requests and added some valuable information about the different aspects pointed out during the first revision, improving considerably the quality of the manuscript. They have also answered with useful explanations the questions included in my comments. I am very happy with the result of these changes and the inclusion of one figure clarifying the mechanistic point of view of the work. However, there is still a minor remaining aspect that I would like to point out before the manuscript will be ready for publication:

- Line 405-407. Please, clarify that when authors said “In this study, we found that the binding of miR-17-5p with its target 3’UTR  mRNA may play a role in maintaining its stability” is not because the results showed in the manuscript lead to that, it is just an hypothesis that authors have after the study. The results shows some correlations that could make you generate that hypothesis but itself, there is no evidence related to the stability. So please, indicate that after the study and “taken together these results with previous publications, a possible explanation could be…”.

Author Response

Reviewer 3

The authors have completed the requests and added some valuable information about the different aspects pointed out during the first revision, improving considerably the quality of the manuscript. They have also answered with useful explanations the questions included in my comments. I am very happy with the result of these changes and the inclusion of one figure clarifying the mechanistic point of view of the work. However, there is still a minor remaining aspect that I would like to point out before the manuscript will be ready for publication:

- Line 405-407. Please, clarify that when authors said “In this study, we found that the binding of miR-17-5p with its target 3’UTR mRNA may play a role in maintaining its stability” is not because the results showed in the manuscript lead to that, it is just an hypothesis that authors have after the study. The results shows some correlations that could make you generate that hypothesis but itself, there is no evidence related to the stability. So please, indicate that after the study and “taken together these results with previous publications, a possible explanation could be…”.

Response: Thank you for the suggestion. We have modified the sentence to become “In this study, we found that miR-17-5p expression levels were also regulated by 3’UTRs of 7 known target mRNAs. Taken together these results with previous publications, a possible explanation is that the binding of miR-17-5p with its target 3’UTR mRNA may play roles in maintaining its stability.”.

Round 3

Reviewer 1 Report

In this updated version, the authors have addressed many of my comments. Of particular interest are the transfection experiments with new 3’UTR sequences (mutated or not). Following the minor comment below, I think the present work is suitable for publication.

l23: correlation in these tissues -> correlation across these tissues